# Cardiac Tomography and Cardiac Magnetic Resonance to Predict the Absence of Intracardiac Thrombus in Anticoagulated Patients Undergoing Atrial Fibrillation Ablation

**DOI:** 10.3390/jcm11082101

**Published:** 2022-04-08

**Authors:** Fatima Zaraket, Deva Bas, Jesus Jimenez, Benjamin Casteigt, Begoña Benito, Julio Martí-Almor, Javi Conejos, Helena Tizón-Marcos, Diana Mojón, Ermengol Vallès

**Affiliations:** 1Electrophysiology Unit, Cardiology Department, Hospital del Mar, 08003 Barcelona, Spain; dbas@psmar.cat (D.B.); jjimenezlopez@psmar.cat (J.J.); bcasteigt@psmar.cat (B.C.); bbenito@psmar.cat (B.B.); jmarti@psmar.cat (J.M.-A.); jconejos@psmar.cat (J.C.); htizon@psmar.cat (H.T.-M.); dmojon@psmar.cat (D.M.); evalles@parcdesalutmar.cat (E.V.); 2Institut Hospital del Mar Investigacions Mèdiques (IMIM), Universitat Autònoma de Barcelona, 08193 Barcelona, Spain

**Keywords:** atrial fibrillation, intracardiac echography, advanced imaging techniques, atrial fibrillation ablation

## Abstract

Background: Pulmonary veins isolation (PVI) is a standard treatment for recurrent atrial fibrillation (AF). Uninterrupted anticoagulation for a minimum of 3 weeks before ablation and exclusion of left atrial (LA) thrombus with transesophageal echography (TEE) immediately before or during the procedure minimize peri-procedural risk. We aimed to demonstrate the utility of cardiac tomography (CT) and cardiac magnetic resonance (CMR) to rule out LA thrombus prior to PVI. Methods: Patients undergoing PVI for recurrent AF were retrospectively evaluated. Only patients that started anticoagulation at least 3 weeks prior to the CT/CMR and subsequently uninterrupted until the ablation procedure were selected. An intracardiac echo (ICE) catheter was used in all patients to evaluate LA thrombus. The results of CT/CMR were compared to ICE imaging. Results: We included 272 consecutive patients averaging 54.5 years (71% male; 30% persistent AF). Average CHA2DS2VASC score was 0.9 ± 0.83 and mean LA diameter was 42 ± 5.7 mm, 111 (41%) patients were on Acenocumarol and 161 (59%) were on direct oral anticoagulants. Anticoagulation was started 227 ± 392 days before the CT/CMR, and 291 ± 416 days before the ablation procedure. CT/CMR diagnosed intracardiac thrombus in two cases, both in the LA appendage. A new CT/CMR revealed resolution of thrombus after six additional months of uninterrupted anticoagulation. No macroscopic thrombus was observed in any patients with ICE (negative predictive value of 100%; *p* < 0.01). Conclusions: CT and MRI are excellent surrogates to TEE and ICE to rule out intracardiac thrombus in patients adequately anticoagulated prior AF ablation. This is true even for delayed procedures as long as anticoagulation is uninterrupted.

## 1. Introduction

Left atrium (LA) catheter ablation with pulmonary vein isolation (PVI) is widely employed as a treatment for patients with symptomatic, drug refractory atrial fibrillation (AF). Transesophageal echocardiography (TEE) is considered the gold standard for detection of LA thrombus prior to catheter ablation with a 97% sensitivity and 100% specificity [1,2,3]. Intracardiac echocardiography (ICE) has also been proved to provide a similar real-time high-resolution imaging capability, which makes it an ideal technique during LA transeptal ablation procedures [4]. On the other hand, computed tomography (CT) and cardiac magnetic resonance (CMR) are commonly utilized to plan procedures by visualizing pulmonary veins and LA anatomy, and there is growing evidence suggesting that these advanced imaging techniques can provide a non-invasive alternative to TEE for evaluation of LA thrombus [5,6]. Guidelines are limited on the precise role of advanced imaging techniques for thrombus evaluation prior to LA ablation [7].

We analyzed LA findings among patients with AF undergoing a CT/CMR study for preprocedure evaluation for PVI, and compared them to the intraprocedural ICE evaluation of the LA. Our hypothesis was that there was correlation between the findings of the CT/CMR and the intraprocedural ICE. The aim of our study was to correlate the CT/MR and intraprocedural ICE findings and determine whether CT/CMR could reduce the need for TEE and/or ICE prior to, or during, catheter ablation of AF.

## 2. Methods

### 2.1. Population

Consecutive patients undergoing recurrent PVI from 2012 to 2017, and with preprocedure CT/CMR were retrospectively included. Only patients with anticoagulation starting at least 3 weeks before CT/CMR and which was uninterrupted until the ablation procedure was selected. All patients were anticoagulated, pre- and postprocedure, according to guidelines. Patients were not anticoagulated after the blanking period of the procedure, except for those with a CHADSVASC score greater than 2. In all patients, an ICE catheter was used to rule out the presence of LA thrombus before transeptal puncture, and also for its guidance. We performed a pre-procedural advanced imaging test (CT/CMR) in all patients submitting to an atrial fibrillation ablation procedure to better understand the individual anatomy and to plan the procedure for each patient (LA size/morphology and PV anatomy variations). The study was conducted in compliance with the most recent version of the Declaration of Helsinki, as well as Spanish laws and regulations (Royal Decree 1090/2015, Royal Decree 1616/2009, Order SAS/3470/2009 of 16 December). The study was assessed and approved by the Ethical Committee of Hospital de Mar, Comité Ético de Investigación Clínica del Consorci Mar Parc de Salut de Barcelona (CEIC-Parc De Salut Mar). All patients signed informed consent before inclusion into the registry.

### 2.2. Imaging Protocols

***CT Imaging protocol***. Multi-slice CT and retrospective ECG-gated spiral acquisitions were performed. In order to improve the specificity, a two-phase scan protocol was used, acquiring a second set of images with a short delay after the initial scan. Traditionally the images are captured only few seconds after contrast arrives at the left heart, making it difficult to differentiate thrombus from sluggish flow. The addition of delayed imaging allows a better distinction between these two elements, considering that a filling defect that persists 1 min after contrast injection is more likely to represent thrombus, whereas sluggish flow is more likely to show contrast opacification in delayed images [6]. ECG gating improves imaging quality (see Figure 1).

***CMR Imaging protocol***. A 1.5 Tesla GE system (GE Signa Excite, GE Healthcare, Ireland) coupled to a 6-element body phased array coil was used. Transverse, coronal, and sagittal plane localizing images were acquired using a FIESTA sequence. Cine-CMRI in the 4- and 2-chamber orientations, using steady-state free precession pulse sequences in held-end expiration was used. The temporal resolution of cine steady-state free precession images was 30 to 40 ms, and was adjusted according to the patient’s heart rate and ability to hold their breath. Late gadolinium enhancement images were obtained after intravenous injection of 0.2 mmol/kg of a gadolinium chelate contrast agent (Dotarem, Gd-DOTA, Guerbet, Villepinte, France).

***ICE Imaging protocol***. The test was performed using a Sequoia ultrasound system with a ViewFlex Xtra ICE catheter (Abbott). During each procedure, the ICE catheter was inserted into the left femoral vein through a 10F introducer sheath and advanced into the right atrium (RA). ICE catheter manipulation included rotation, advancement, withdrawal, and the possibility of antero-posterior and right to left deflections using two jog dials. Once in the RA, via gradual clockwise rotation of the catheter, we could obtain panoramic views of the interatrial septum, LA, mitral annulus, LA appendage, and PVs. ICE was used to rule out the presence of thrombus in any chamber, including the LA appendage (see Figure 2). Subsequently, it was used to identify the fossa ovalis to guide the transeptal puncture and aid in positioning the ablation catheter in the pulmonary vein antrum. At our center, approximately 100 transeptal punctures have been performed every year since 2009, always under ICE guidance. Thus, operators have developed high cognitive and technical skills concerning this advanced technique.

### 2.3. Procedural Characteristics

Briefly, a single trans-septal puncture is performed using a standard long sheath (Lamp, St. Jude Medical, St. Paul, MN, USA), guided by intracardiac echo (ViewFlex, St. Jude Medical) with the patient under conscious sedation and under infusion of unfractionated heparin aiming at an activated clotting time (ACT) of 250–300. The transeptal sheath is exchanged over a guidewire for a 15F deflectable introducer (Flexcath, Medtronic, Minneapolis, MN, USA), and a CB2 (Arctic Front Advance, Medtronic) is introduced, together with an inner circular mapping catheter (Achieve, Medtronic), into the antrum of each PV. All procedures were performed under the guidance of a three-dimensional reconstruction of the left atrium and the pulmonary veins, extracted from a cardiac CT or MRI scan, and using an esophageal temperature probe with multiple thermocouples and adjustments to the balloon position. Cryoenergy delivery was stopped in case of temperature drop below 26 °C. The right phrenic nerve was monitored with palpation and fluoroscopy during pacing. PV isolation was defined by the persistent elimination or dissociation of PV potentials, visualized by the circular mapping catheter (Achieve catheter). Time-to-effect (TTE) was defined as the time to PV isolation during ablation.

### 2.4. Statistical Analysis

Continuous variables were presented as mean ± standard deviation or median. The categorical data were expressed as frequencies and percentages. Univariate comparisons were performed using the χ^2^ test for categorical variables. A *p* value of <0.05 was considered statistically significant. All analyses were performed using SPSS for Mac, version 20.0 (SPSS Inc., Chicago, IL, USA).

## 3. Results

### 3.1. Population Characteristics

We included 272 consecutive patients, averaging 54.5 ± 4.94 years (71% male). Mean AF evolution time was 3.3 years. The population was relatively healthy with a CHA2DS2VASC below 2 in 72% of cases. The average CHA2DS2VASC score was 0.9 ± 0.83 and the LA diameter was 42 ± 5.7 mm. Up to 22% of patients had an anatomic variation of PV, mainly a common left ostium (see Table 1).

Approximately, in 21 patients (7%), AF was recorded as heart rhythm at the time of CMR scanning, but all with a controlled heart rate, allowing a good-quality test.

### 3.2. Anticoagulation Therapy

Of the 272 patients, 111 were on Acenocumarol (41%), and 161 (59%) patients were on direct oral anticoagulants: 38 patients were on Dabigatran (14%), 55 were on Rivaroxaban (20%), 63 were on Apixaban (23%), and 5 were on Edoxaban (2%). Anticoagulation was started 227 ± 392 days before CT/CMR, and 291 ± 416 days before the ablation procedure (see Table 2); it was not discontinued during ablation. Mean basal ACT, just before the daily anticoagulation dose given prior to the procedure was 162.2 ± 32.6, and mean ACT during the procedure was 310.4 ± 73.4.

### 3.3. Thrombus Detection

CT/CMR was diagnostic of intracardiac thrombus in only 2 out of the 272 cases (0.7%; both within the LA appendage). Those two patients, which were on Acenocumarol with a TTR of 70% (INR at the moment of thrombus detection: 2.1 and 2.2, respectively), remained on anticoagulants for an additional 6 month period and had a higher INR goal and increased INR controls, after which CT/CMR was negative for intracardiac thrombus. Time between CT/CMR and the procedure was 64 ± 142 days. At the time of the procedure no macroscopic thrombus was observed in any patients with ICE. Therefore CT/CMR achieved a negative predictive value of 100% compared to ICE (*p* < 0.01).

### 3.4. Cost Analyses

From an economic point of view, in the setting a public health system, the cost of the use of CT or CMR prior to AF ablation procedures to rule out intracardiac thrombus needs to be weighed against the cost of use of TEE or ICE. The cost of a CT scan or a CMR in our hospital is approximately 250 euros, while the cost of an ICE catheter is 1900 euros. Regarding TEE, CT/CMR has the advantages of being noninvasive, having a shorter procedural time, does not need anesthesia support, and provides left atrial and pulmonary vein structure, which is useful for procedural planning.

### 3.5. Follow-Up

Patients had follow-up at 1 and 6 months after the procedure, and every 6 months thereafter. A 12-lead ECG and a 24-h Holter monitor were used during the first visit and an additional 3 or 7 days of Holter monitoring was performed if the patients referred to symptoms, or in case of frequent supraventricular ectopy in the ECG. Furthermore, patients were instructed to visit emergency services whenever they experienced symptoms suggestive of arrhythmia relapses. Antiarrhythmic drugs were continued during the 3 month blanking period, but were withdrawn if there were no AF recurrences. Continuation of oral anticoagulation was indicated according to the CHADSVASC score. Only one patient experienced a catheter ablation-related stroke (0.3%), defined as occurring within the first month after the procedure. This patient received appropriate oral anticoagulation therapy (Acenocumarol) before the procedure, had a TTR of 80% and INR controls every 4 weeks, starting 1537 days before the ablation, and was uninterrupted after the procedure, since the CHA2DSVAS2C score was 3. The ACT for this patient had been >250 s during the procedure, which had been particularly long and complex due to difficulties in isolating the left superior PV. INR the day of the procedure was 2.2. No thrombus was observed either with a CT scan performed 32 days before the ablation, nor with the periprocedural ICE.

## 4. Discussion

Catheter ablation has been established as a standard treatment for symptomatic AF, and PVI has become the cornerstone of such treatment and is recommended in the current guidelines [8]. Several studies have shown that advanced cardiac imaging techniques can improve AF ablation results [9,10]. Both CT scan and CMR provide accurate anatomic details (i.e., number and size of the PVs and the LA, and anatomic variants of the PVs) in a non-invasive manner, and can help an electrophysiologist to plan the procedure. Furthermore, advanced imaging techniques can be integrated into the 3D navigation systems used to perform AF ablation procedures, which can provide real-time non-fluoroscopic intraprocedural navigation guidance [11,12,13]. Patients with AF have increased risk of stroke from left atrial thrombi. Consequently, the need to rule out intracardiac thrombus is mandatory prior to the ablation procedure. Historically, TEE has been considered the “gold standard” imaging technique in order to rule out left atrial thrombus, and it is usually performed 24 to 48 h before the procedure. However, TEE is an invasive procedure that requires sedation and sometimes intubation. Therefore, other alternatives have been investigated: ICE can both rule out the presence of thrombus and guide the procedure [14,15], but at a higher cost. Despite the non-negligible economic aspect, ICE presents important advantages, allowing the direct visualization of the endocardium and locating the needle and the sheath exactly against the interatrial septum. It can be even more useful to allow a safe puncture, especially in cases of thickened or floppy septums, interatrial septal aneurysm, lipomatous hypertrophy, previous suture of the septum after cardiac surgery, or in the presence of additional fibrosis after previous procedures, avoiding patient discomfort due to the introduction of the TEE probe and general anesthesia required for the latter. ICE catheter manipulation can offer an anterior/posterior orientation, showing, at the same time, the aorta and the interatrial septum, and offers the possibility of displaying and examining pulmonary veins, the eventual presence of a common trunk, guaranteeing extra support in all the case of difficult transeptal puncture. This important tool can, not only exclude the presence of LAA thrombus, but also assess its size, form, and anatomical relationship with the pulmonary veins. Furthermore, it allows early detection of intra-procedure complications, such as pericardial effusion or aortic root, thanks to the direct visualization of these structures.

On the other hand, other advanced imaging techniques, such CT scan or CMR, have been increasingly used in current clinical practice, and may be able to replace TEE or ICE in a non-invasive manner.

Several studies investigated the use of CT scan and CMR for the assessment of LA thrombus in pre-procedural evaluation of AF ablation. A metanalysis from Romero et al., showed an overall high accuracy of CT scan compared to TEE for the detection of LA/LAA thrombus. Other metanalyses of prospective studies comparing CT with TEE have had similar results, and support the use of CT to rule out LA thrombus, especially when delayed imaging, ECG gating, and heart rate control are performed [6,16]. Similar evidence has been demonstrated for CMR as well, which is another noninvasive technique that has been shown to be as effective as TEE in evaluating for LA thrombus, in addition to providing LA and PV anatomy and structure [10,17,18]. We have to keep in mind that these techniques are not without limits, such as, for CT, the risk of contrast-induced nephropathy, even if is relatively low in patients with normal renal function and additional radiation, even if, nowadays, thanks to the advances in this field and the use of the delayed imaging method, radiation doses are much lower than in earlier studies (<3 millisieverts compared with >15 millisieverts) [19].

None of these studies directly compared advanced imaging techniques to ICE. The purpose of our study was to examine the cost-effectiveness and time-efficiency advantages of pre-procedural CT scan or CMR, performed remotely (in our case an average of two months pre-procedure), compared to the use of TEE or ICE to evaluate for LA thrombus prior to AF ablation. We found this strategy to be a safe, non-invasive, and cost-effective alternative for LA thrombus detection. Importantly, this was true even for deferred procedures, as long as the anticoagulation therapy continued uninterrupted. To our knowledge, this is the largest study comparing the accuracy of LA thrombus detection with non-invasive advanced image techniques compared to periprocedural ICE [20,21].

In conclusion, it seems clear that employing pre-procedural CT/CMR as the only strategy to rule out LA/LAA thrombus is efficient and safe, as well as non-invasive. It also appears, as previously described, that the use of CT/CMR can limit procedural costs while preserving high-quality care [22].

## 5. Limitations

There were several limitations in our study, including a low, but anticipated, prevalence of LAA thrombi in this selected study population undergoing atrial fibrillation ablation. This can be attributable to low-risk patients, even if is in keeping with the relatively low prevalence of LAA thrombus, as reported in the literature in a similar cohort [23]. The aim of our study is to demonstrate that CMR and CT are highly specific in ruling out LAA thrombus in the group of patients most important to exclude. Combined tools can be used in patients with a higher risk. The results of our study reflect the experience from a single center. The obtained results were derived from a retrospective analysis with a low sample size.

## 6. Conclusions

CT and CMR are excellent surrogates of TEE or ICE to rule out intracardiac thrombus in patients adequately anticoagulated prior to AF ablation. This is true even for deferred procedures, as long as anticoagulation is uninterrupted. Furthermore, pre-procedural imaging provides key anatomic information that can be complemented by intraprocedural electroanatomic mapping to minimize procedural complications. Further multicenter studies with larger sample sizes are necessary to confirm our findings.

## Figures and Tables

**Figure 1 jcm-11-02101-f001:**
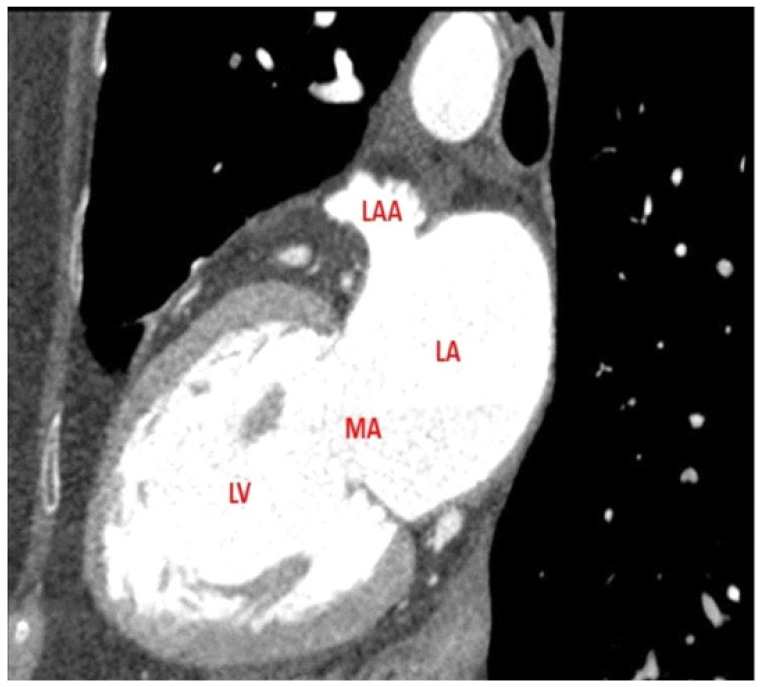
CT scan caption showing the absence of thrombus in the LA, including the LAA. LA: left atrium; LAA: left atrium appendage; LV: left ventricle; MA: mitral annulus.

**Figure 2 jcm-11-02101-f002:**
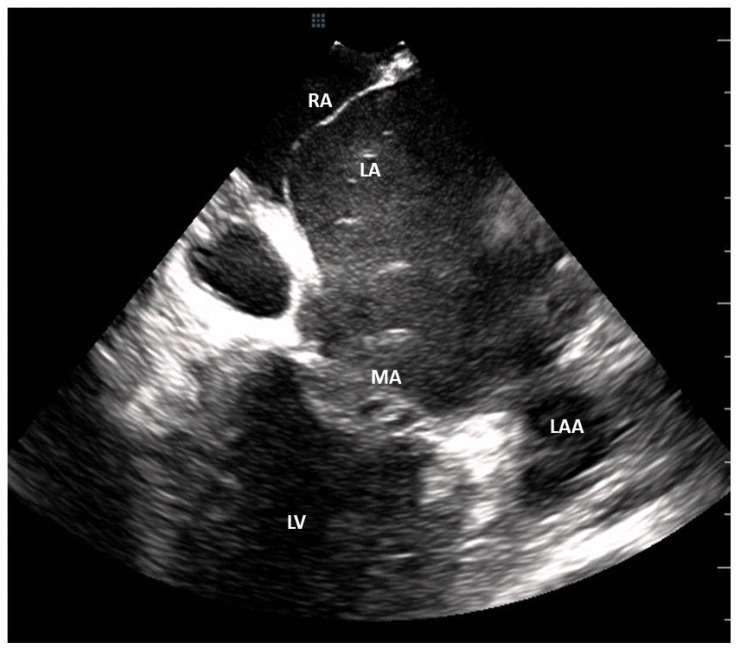
ICE caption with the transducer placed into the low right atrium, where absence of thrombus can be observed in the LA, including the LAA. LA: left atrium; LAA: left atrium appendage; LV: left ventricle; MA: mitral annulus; RA: right atrium.

**Table 1 jcm-11-02101-t001:** Population characteristics.

	Patients = 272
Age: years (Ave ± st dev)	54.5 ± 4.94
Gender, male: *n* (%)	194 (71)
Hypertension: *n* (%)	109 (40)
CHA2DS2VASC score < 2	197 (72)
CHA2DS2VASC (Ave ± st dev)	0.9 ± 0.83
Type, persistent AF: *n* (%)	38 (14)
AF evolution time: months (Ave ± st dev)	40 ± 41.4
LV ejection fraction: (Ave ± st dev)	59 ± 9.3
LA diameter: mm (Ave ± st dev)	42 ± 5.7
Image tecnique: *n* (%) CT scan	147 (54)
CMR	125 (46)
PV anatomic variation: *n* (%)	55 (20)

AF: atrial fibrillation; Ave: average; CMR: cardiac magnetic resonance; CT: computed tomography; LA: left atrium; LV: left ventricle; PV: pulmonary veins; st dev: standard deviation.

**Table 2 jcm-11-02101-t002:** Treatment and procedural characteristics.

	Total Patients = 272
Time from AC to ablation: days (Ave ± st dev)	227 ± 392
Time from CT/CMR to ablation: days (Ave ± st dev)	291 ± 416
Type of ablation: *n* (%) RF ablation	89 (33)
Cryoballoon ablation	183 (67)
Anticoagulant drugs: *n* (%)	161(59)
Dabigratan	38 (14)
Rivaroxaban	55 (20)
Apixaban	63 (23)
Edoxaban	5 (2)
Acenocumarol	111 (41)
Basal ACT: seconds (Ave ± st dev)	162.2 ± 32.6
30 min ACT: seconds (Ave ± st dev)	310.4 ± 73.4

AC: anticoagulation; ACT: activated clotting time; Ave: average; CMR: cardiac magnetic resonance; CT: computed tomography; RF: radiofrequency; st dev: standard deviation.

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
