# Peer review of "Cardiac Tomography and Cardiac Magnetic Resonance to Predict the Absence of Intracardiac Thrombus in Anticoagulated Patients Undergoing Atrial Fibrillation Ablation"

_jcm, 2022, doi:10.3390/jcm11082101_

Round 1

Reviewer 1 Report

Dear Editor dear Authors, thank you for the opportunity to review the article „Cardiac tomography and cardiac magnetic resonance to predict the absence of intracardiac thrombus in anticoagulated patients undergoing atrial fibrillation ablation“

In general I found the article interesting, but I have some comments:

The topic is indeed interesting and the results presented in this study can have an important role for further development of this direction. However, presented patients cohort with CHADsVASc score below 2 Points in more than 2/3 cases suggests fewer cases of thrombus formation by itself.

CMR gaiting happens with ECG triggering and to the best of my knowledge serious problems for assessment of the thrombus can occur once AF is presented. Please comment this in Discussion section.

Please provide information about the SR vs AF in patients in CMR group in result section.

Computed tomography is cheap and fast, but we all forget about radiation exposure, which is very important. Please comment on this topic in the Discussion section.

Best Regards

Reviewer 2 Report

Zaraket and colleagues present a single-center retrospective study on cardiac CT and cardiac MRI performed for rule out of intracardiac thrombi in patients prior to atrial fibrillation ablation.

The manuscript is comprehensive and structured. Nevertheless, there are several major limitations:

  • The study population is relatively small with only 272 enrolled patients. Furthermore, patients were only enrolled retrospectively with typical limitations of data acquisition and quality.
  • The authors report that only two patients suffered from ischemic stroke during follow-up. Unfortunately, the authors do not state how long the follow-up was, how many patients underwent follow-up and how follow-up was performed. This is inacceptable.
  • Throughout the whole manuscript the fact that CT-scans for thrombus rule out represents additional radiation has not been discussed. In fact, the authors describe how two patients with CT thrombus detection underwent a second CT-scan for rule out. The authors definitely have to address the topic of additional radiation.
  • In the abstract section the average CHA2DS2-VASC score was calculated to be 0.9. Seemingly this is a medium. In cases of medium the authors should give the standard derivation. However, CHA2DS-VASC score represents a categorical value and therefore should be presented as median and interquartile range.
  • The lack of presenting standard deviations can be found throughout the whole manuscript and is inacceptable.
  • In patients on vitamin k-antagonist therapy no data on INR at timepoint of thrombus rule out are present. The authors should comment on this.
  • The manuscript does not contain a limitations section. This is scientific standard and needs to be added.
  • The authors do not describe the standard of periprocedural management in their hospital. In the methods section it should be described, why patients with > 3 weeks oral anticoagulation underwent again imaging for rule out of cardiac thrombi
  • The mean ChA2DS2-VASC-Score was 0.9 and therefore was very low (beneath indication for permanent oral anticoagulation). Taking this into account the population does not fit to the aim of this study.
  • The manuscript has many spelling mistakes that should be corrected.

Reviewer 3 Report

-  I don’t have any comments on methods, data description and conclusion

-  The authors didn’t mention that intracardiac echo is really useful not only for thrombus detection but also for transeptal access. In a “zero fluoro” view of AF ablation ICE is crucial. ICE gives also real time info on esophageal position (mobile structure). Maybe these aspects could be discussed.
